# Association of *MGMT* Promoter and Enhancer Methylation with Genetic Variants, Clinical Parameters, and Demographic Characteristics in Glioblastoma

**DOI:** 10.3390/cancers15245777

**Published:** 2023-12-09

**Authors:** Katja Zappe, Katharina Pühringer, Simon Pflug, Daniel Berger, Serge Weis, Sabine Spiegl-Kreinecker, Margit Cichna-Markl

**Affiliations:** 1Department of Analytical Chemistry, Faculty of Chemistry, University of Vienna, 1090 Vienna, Austria; katja.zappe@univie.ac.at (K.Z.); katharina.puehringer@univie.ac.at (K.P.); n12030136@students.meduniwien.ac.at (S.P.); office@berger-daniel.at (D.B.); 2Division of Neuropathology, Department of Pathology and Molecular Pathology, Kepler University Hospital GmbH, Johannes Kepler University, 4040 Linz, Austria; serge.weis@kepleruniklinikum.at; 3Department of Neurosurgery, Kepler University Hospital GmbH, Johannes Kepler University, 4040 Linz, Austria; sabine.spiegl-kreinecker@kepleruniklinikum.at

**Keywords:** glioblastoma, MGMT, DNA methylation, enhancer methylation, promoter methylation, biomarker, Ki-67, overall survival, age

## Abstract

**Simple Summary:**

Glioblastoma is the most common and most aggressive brain tumor in adults. The chemotherapeutic substance temozolomide plays an important role in glioblastoma therapy. However, the response of glioblastoma patients to temozolomide depends on the expression level of the repair protein O6-methylguanine-DNA methyltransferase (MGMT) in the tumor. Since the DNA methylation status of the MGMT promoter has an impact on MGMT expression, it serves as biomarker for predicting the response to temozolomide and prognosis of glioblastoma. We investigated if the DNA methylation status of enhancers, other important regulatory elements, is also associated with MGMT expression. We found the methylation status of several regions of enhancers to be associated with MGMT expression. In addition, we found associations with common genetic variants and clinical parameters, including overall survival.

**Abstract:**

The response of glioblastoma (GBM) patients to the alkylating agent temozolomide (TMZ) vitally depends on the expression level of the repair protein O6-methylguanine-DNA methyltransferase (MGMT). Since MGMT is strongly regulated by promoter methylation, the methylation status of the *MGMT* promoter has emerged as a prognostic and predictive biomarker for GBM patients. By determining the methylation levels of the four enhancers located within or close to the *MGMT* gene, we recently found that enhancer methylation contributes to *MGMT* regulation. In this study, we investigated if methylation of the four enhancers is associated with SNP rs16906252, *TERT* promoter mutations C228T and C250T, *TERT* SNP rs2853669, proliferation index Ki-67, overall survival (OS), age, and sex of the patients. In general, associations with genetic variants, clinical parameters, and demographic characteristics were caused by a complex interplay of multiple CpGs in the *MGMT* promoter and of multiple CpGs in enhancer regions. The observed associations for intragenic enhancer 4, located in intron 2 of *MGMT*, differed from associations observed for the three intergenic enhancers. Some findings were restricted to subgroups of samples with either methylated or unmethylated *MGMT* promoters, underpinning the relevance of the *MGMT* promoter status in GBMs.

## 1. Introduction

According to the recent World Health Organization (WHO) classification of tumors of the central nervous system, glioblastomas (GBMs) are isocitrate dehydrogenase (IDH)-wildtype diffuse astrocytomas WHO grade 4 [1]. GBMs are characterized by high aggressiveness, poor prognosis, and limited treatment options. Standard therapy consists of radiotherapy plus concomitant and adjuvant therapy with the alkylating agent temozolomide (TMZ) [2]. However, response to TMZ vitally depends on the expression level of O6-methylguanine-DNA methyltransferase (MGMT), a protein repairing DNA damage by transferring the alkyl group from DNA to a cysteine residue in its active center [3]. Tumor cells lacking MGMT expression are frequently sensitive to TMZ, whereas MGMT-expressing tumors are commonly resistant to alkylating agents, including TMZ [4].

DNA methylation plays a crucial role in *MGMT* regulation. Analogous to various other genes containing a cytosine guanine dinucleotide (CpG) island in their promoter, methylation of the *MGMT* promoter results in transcriptional silencing [5,6]. Several studies indicate that *MGMT* promoter methylation confers a survival benefit to GBM patients treated with TMZ [7,8,9]. Thus, the *MGMT* promoter methylation status has emerged as a predictive biomarker for the response to TMZ, in particular in newly diagnosed patients of higher age [10]. In addition, *MGMT* promoter methylation is considered a suitable prognostic biomarker for GBM [11,12].

However, growing evidence suggests that not all tumors with an unmethylated *MGMT* promoter express MGMT, whereas in other tumors, MGMT expression was detected in spite of *MGMT* promoter methylation [13]. In a recent study, we investigated if enhancer methylation is involved in *MGMT* regulation [14]. We selected four enhancers that had previously been associated with *MGMT*, including three intergenic enhancers (“enhancer 1”, hs737, [15]; “enhancer 2”, identified by Chen et al. [16]; “enhancer 3” (hs699, [15]) and one intragenic enhancer located in *MGMT* intron 2 (“enhancer 4”, hs696, [15]). For DNA methylation analysis, we developed assays based on polymerase chain reaction (PCR) and pyrosequencing (PSQ), targeting eight out of 27 CpGs of enhancer 1, 19 out of 46 CpGs of enhancer 2, eight out of 33 CpGs of enhancer 3, and 14 out of 26 CpGs of enhancer 4. Application of the assays to primary human tumor cell lines derived from GBM patients revealed that the DNA methylation status of *MGMT* enhancers 2, 3, and 4 is associated with *MGMT* promoter methylation and/or MGMT protein expression. In addition, we found the methylation status of two CpG regions of enhancer 4 to be associated with the overall survival (OS) of GBM patients, suggesting that enhancer methylation is a potential prognostic biomarker for GBM [14].

In this study, we investigated if *MGMT* enhancer methylation is associated with common genetic variants in GBM patients, e.g., *MGMT* SNP rs16906252 C>T and telomerase reverse transcriptase (TERT) promoter mutations C228T and C250T, clinical parameters, e.g., proliferation index Ki-67, postoperative Karnofsky Performance Score (KPS), progression-free survival (PFS), and overall survival (OS), as well as age and sex of the patients.

## 2. Materials and Methods

### 2.1. Samples and Cell Culturing

The sample set consisted of primary human tumor cell lines derived from 38 GBM patients who underwent surgery between 2001 and 2020 at the Department of Neurosurgery, Kepler University Hospital GmbH, Johannes Kepler University Linz. Primary cell lines were established as described previously [17]. For 35 patients (GBM01-GBM35), stable cell cultures were obtained, whereas GBM derived cell cultures for three patients (GBM36–GBM38) did not develop into stable cell lines but ceased growth after several passages. The study was approved by the local Ethics Commission of the Faculty of Medicine at the Johannes Kepler University Linz (application number E-39-15). Informed written consent was obtained from all patients.

To closely mirror the original tumor, all cell cultures were used at passages between 2 and 6. Cell lines were cultured in RPMI-1640, 7% fetal calf serum (FCS), and 1% glutamine without antibiotics (all Sigma-Aldrich, Schnelldorf, Germany) at 37 °C in a humidified 5% CO_2_ incubator. Cells were harvested before reaching the confluence between passages 2 and 6 and pelleted by centrifugation. Cell pellets were stored at −80 °C until DNA extraction.

### 2.2. DNA Methylation Analysis

DNA methylation analysis was performed previously [14]. In brief, genomic DNA was isolated using the QIAamp DNA Blood Mini Kit (Qiagen, Hilden, Germany), and bisulfite was converted using the EpiTect Fast Bisulfite Conversion Kit (Qiagen, Germany). After amplifying the target regions by PCR, the DNA methylation status of individual CpGs was determined by PSQ using the PyroMark Q24 Vacuum Workstation, PyroMark Q24 Advanced instrument with PyroMark Q24 Advanced Accessories, and PyroMark Q24 Advanced CpG Reagents (all Qiagen).

The numbering of the 98 CpGs in the *MGMT* promoter was done according to the CpG island given in the University of California at Santa Cruz Genome (UCSC) Browser (GRCh38/hg38) [18] as suggested by Wick et al. [19].

### 2.3. Determination of Genetic Variants and Clinical Parameters

*TERT* promoter mutations C228T, C229A, and C250T, as well as SNP rs2853669 T>C>G genotypes, were determined previously by sequencing using BigDye Terminator v1.1 Cycle Sequencing Kit (Applied Biosystems, Waltham, MA, USA) and a 3130 Genetic Analyzer (Applied Biosystems) following standard procedures [20]. Genotypes for *MGMT* SNP rs16906252 C>T were determined previously for samples GBM01–GBM18 [21]; the genotyping of samples GBM19–GBM38 was done in this study by high-resolution melting (HRM) coupled with PSQ using the “3 SNPs assay” as described by Zappe et al. [21].

Relative MGMT protein expression (related to cell line GL80; expression in GL80 was set as 1) was determined by Western blot analysis [20].

Corresponding clinical parameters (age, sex, therapy, KPS, OS) were available for all patients. OS was defined as the period between the date of surgery and death. Progression-free survival (PFS), defined as the time between diagnosis and disease progression, was only available for 20 patients. In addition, the proliferation index Ki-67 was assessed during routine histopathologic diagnostics by immunohistochemistry.

### 2.4. Data Analysis and Statistics

Data were analyzed and presented graphically using R version 3.6.2 [22]. A Student’s *t*-test and one-way ANOVA (analysis of variance) followed by a post-hoc *t*-test corrected for multiple testing by Holm’s *p*-value adjustment was applied to test for significant differences between groups. Groups consisting of only one member were excluded from testing. Effects of multiple variables were analyzed by two-way ANOVA. Scatterplots and Pearson’s correlation coefficients were used to assess the relationship between two quantitative variables. The Kaplan–Meier estimator and Cox proportional hazards models with log-rank tests were used to analyze survival data.

Data obtained for unstable cell cultures (GBM36–GBM38) were excluded from statistical analysis and are discussed separately (Section 3.14).

## 3. Results

### 3.1. Clinical Parameters and Demographic Characteristics of the Patient Cohort

Clinical characteristics of the 35 GBM patients for which stable cell cultures could be established are summarized in Table 1. The age of patients at surgery ranged from 44 years to 85 years (mean 62.2 years, median 64.0 years). The Ki-67 index, specified for 15 GBM patients, ranged from 15% to 90%. For a further nine patients, the Ki-67 status was available semi-quantitatively. Thirteen (54.2%) patients showed a Ki-67 index > 50%. Twenty-one GBM patients underwent radio-chemotherapy, seven radiotherapy, one patient received chemotherapy only, and seven patients did not receive any primary therapy after surgery. Fourteen (40.0%) of the 35 GBM patients were females, 21 (60.0%) were males. The median PFS was 5.26 months. KPS ranged from 40 to 100%. OS was in the range of 0.89 months to 52.50 months (mean 15.31 months, median 11.15 months).

Clinical and demographic data of GBM patients for which stable cell cultures could not be established are summarized in Appendix A.

### 3.2. Methylation Status of the MGMT Promoter and Enhancers for the Patient Cohort

DNA methylation levels determined for the patient cohort were published recently [14]. DNA methylation data were provided for 49 CpGs in *MGMT* enhancers, including eight CpGs (CpGs 12–19) of enhancer 1 (hs737, [15]), 19 CpGs (CpGs 05–08, CpGs 11–18, CpGs 24–27, and CpGs 37–39) of enhancer 2 (identified by Chen et al. [16]), eight CpGs (CpGs 15–22) of enhancer 3 (hs699, [15]), and 14 CpGs (CpGs 01–03, 07–08, 09–13, and 19–22) of enhancer 4 (hs696, [15]). In addition, the paper provided the methylation status of 12 CpGs (CpGs 72–83) in the *MGMT* promoter [14].

### 3.3. Association between MGMT Promoter/Enhancer Methylation and MGMT SNP rs16906252 C>T

By genotyping *MGMT* SNP rs16906252, 32 (91.4%) GBM samples were found to be homozygous C (wildtype), two (5.7%) samples (GBM20 and GBM27) were homozygous T, and for one (2.9%) sample (GBM06), an (atypical) heterozygous genotype was determined. Homozygous C and homozygous T samples did not differ (*p* ≥ 0.269) in their *MGMT* promoter methylation status (Figure 1a). In addition, wildtype samples and homozygous SNP carriers did not differ (*p* = 0.059) in the methylation status of enhancer 1 if all eight CpGs targeted (CpGs 12–19) were taken into account (Figure 1b). However, individual CpGs 13, 14, and 15 of enhancer 1 were significantly more highly methylated in homozygous SNP carriers than in wildtype samples (Figure 1b). In line with enhancer 1, homozygous SNP carriers showed higher methylation status of enhancer 3 compared to wildtype samples (Figure 1c).

Significant differences between wildtype and homozygous SNP carrying samples were also found for CpGs in enhancer 2 (Figure 1d) and enhancer 4 (Figure 1e). However, in contrast to enhancer 1 and enhancer 3, methylation levels of enhancer 2 and enhancer 4 methylation were lower in homozygous SNP carriers than in wildtype samples.

### 3.4. Association between MGMT Promoter/Enhancer Methylation and TERT Promoter Mutations C228T and C250T

Only three (8.6%) of the 35 GBM samples were lacking *TERT* promoter mutations. C228T and C250T mutations were detected in 21 (60.0%) and eleven (31.4%) tumors, respectively. None of the GBM samples had the C229A mutation.

In the subgroup of tumors with methylated *MGMT* promoter, C250T mutation carriers showed significantly higher methylation of CpGs 72–83 in the *MGMT* promoter compared to wildtype tumors and C228T mutation carriers (Figure 2a). No significant differences were found for individual CpGs (*p* ≥ 0.198). Since GBM18 was the only sample with an unmethylated *MGMT* promoter lacking *TERT* promoter mutation, it was excluded from statistical analysis.

Methylation levels of enhancer 1, enhancer 2, and enhancer 3 were also associated with *TERT* promoter mutations (Figure 2b–d). For enhancer 4, we did not find any association between the methylation status and *TERT* promoter mutations (*p* ≥ 0.079).

### 3.5. Association between MGMT Promoter/Enhancer Methylation and TERT SNP rs2853669

By genotyping the patient cohort for *TERT* SNP rs2853669, 19 (55.9%) GBM samples were found to have the wildtype TT genotype and 15 (44.1%) were heterozygous CT. The homozygous CC genotype was not detected in any samples.

GBM patients of the CT genotype had a significantly lower *MGMT* promoter methylation status (CpG 72–83) than those of the TT genotype (Figure 3a). For individual CpG, no significant differences were found (*p* ≥ 0.090).

Significant differences were also found for the four enhancers investigated (Figure 3b–e). The three intergenic enhancers were found to be more highly methylated in samples of the CT genotype compared to wildtype samples. In contrast, enhancer 4 was significantly lower methylated in samples of the CT genotype compared to those of the TT genotype. The difference was predominantly observed for samples with an unmethylated *MGMT* promoter. In samples with a methylated *MGMT* promoter, individual CpG 09 was lower methylated in heterozygous samples than in those showing the wildtype (Figure 3e).

### 3.6. Association between MGMT Promoter/Enhancer Methylation and Proliferation Index Ki-67

We did not find a significant correlation between the Ki-67 index and the methylation status of the *MGMT* promoter (*p* ≥ 0.067), enhancer 3 (*p* ≥ 0.073) or enhancer 4 (*p* ≥ 0.094), independent if GBM samples were stratified by their *MGMT* promoter methylation status or not. However, we found the Ki-67 index to be associated with the methylation status of individual CpGs of enhancer 1 and enhancer 2. Three CpGs of enhancer 1 (CpG 12 (*r* = 0.84, *p* = 0.019), CpG 13 (*r* = 0.76, *p* = 0.047), and CpG 18 (*r* = 0.83, *p* = 0.020)) and ten CpGs of enhancer 2 (CpG 05 (*r* = 0.90, *p* = 0.005), CpG 07 (*r* = 0.93, *p* = 0.002), CpG 08, (*r* = 0.95, *p* = 0.001), CpG 11 (*r* = 0.92, *p* = 0.004), CpG 12 (*r* = 0.85, *p* = 0.015), CpG 14 (*r* = 0.77, *p* = 0.043), CpG 15 (*r* = 0.80, *p* = 0.032), CpG 18 (*r* = 0.80, *p* = 0.030), CpG 37 (*r* = 0.97, *p* = 0.002), and CpG 38 (*r* = 0.96, *p* = 0.002)) showed a significantly positive correlation with Ki-67 (Figure 4a). In the case of enhancer 2, mean methylation levels of CpGs 05–08 (*r* = 0.94, *p* = 0.002), CpGs 11–14 (*r* = 0.82, *p* = 0.023), CpGs 15–18 (*r* = 0.79, *p* = 0.035), and CpGs 37–39 (*r* = 0.94, *p* = 0.005) also significantly correlated with the Ki-67 index. Interestingly, these correlations were exclusively found for GBM samples with an unmethylated *MGMT* promoter, as shown for the CpG 18 of enhancer 1 (Figure 4b) and for the CpG 08 of enhancer 2 (Figure 4c).

Next, we investigated if GBM samples stratified by their Ki-67 index differed in *MGMT* promoter and/or enhancer methylation. By setting a cut-off of 50%, we found significant differences between GBM samples with high (>50%) and those with low (≤50%) Ki-67 index for the *MGMT* promoter but also for enhancer regions (Figure 4d–g). Methylation levels of CpGs 72–83 in the *MGMT* promoter (Figure 4d), CpGs 12–19 of enhancer 1 (Figure 4e), and CpGs 05–08 and CpGs 37–39 of enhancer 2 (Figure 4f) were significantly higher in GBM patients with high Ki-67 index (>50%) compared to those with low (≤50%) Ki-67 index. The difference was also significant for individual CpG 08 of enhancer 2 (Figure 4f). In samples with an unmethylated *MGMT* promoter, Ki-67 status > 50% was associated with significantly higher methylation of CpGs 05–08, CpGs 24–27, CpGs 37–39, mean of CpGs 37–39, and individual CpG 08 and CpG 37 (Figure 4f). In GBM samples with methylated *MGMT* promoter, methylation of CpGs 15–22 of enhancer 3 was higher in patients with Ki-67 > 50% (Figure 4g).

Among all CpGs targeted, only CpGs 15–18 of enhancer 2 were significantly lower methylated in patients with high Ki-67 status but exclusively in samples with methylated *MGMT* promoter (Figure 4f).

### 3.7. Association between MGMT Promoter/Enhancer Methylation and KPS

We did not find *MGMT* promoter or enhancer methylation to be correlated with KPS in our patient cohort (*p* ≥ 0.070).

### 3.8. Association between MGMT Promoter/Enhancer Methylation and PFS

Neither *MGMT* promoter nor enhancer methylation was correlated with PFS if GBM samples were not stratified by their *MGMT* promoter methylation status (*p* > 0.087). However, in samples with an unmethylated *MGMT* promoter, methylation of CpG 01 in enhancer 4 significantly negatively correlated with PFS (*r* = −0.84, *p* = 0.009).

### 3.9. Association between MGMT Promoter/Enhancer Methylation and OS

By using a cut-off of 8%, patients with methylated *MGMT* promoter had significantly longer OS compared to patients with an unmethylated *MGMT* promoter (Figure 5a).

No significant differences in OS were found for the methylation of enhancer 1 (Figure 5b), enhancer 2 (Figure 5c), and enhancer 3 (Figure 5d). However, for enhancer 4 we found significantly longer OS for patients showing low methylation of CpGs 01–03, CpGs 09–13, and individual CpG 07, but not for individual CpG 08 (Figure 5e–i) and mean methylation of CpGs 07–08 (cut-off 55%, *p* = 0.117).

The hazard ratio for individual CpGs of the *MGMT* promoter and enhancer 4 is given in Table 2. Methylation levels ≥8% of the CpGs targeted in the *MGMT* promoter as well as methylation levels <55% of the mean of CpGs 01–03, mean of CpGs 09–13, and methylation levels of CpG 03, CpG 07, and CpG 13 offer survival benefit for GBM patients.

### 3.10. Associations of Various Clinical Parameters

Surprisingly, for 15 GBM patients for which the Ki-67 index was available quantitatively, the Ki-67 index was significantly positively correlated with OS (r = 0.79, *p* < 0.001) (Figure 6a). GBM patients with a Ki-67 index > 50% had significantly longer OS compared to patients with a low (≤50%) Ki-67 index (Figure 6b, Table 3). The significant positive association between the Ki-67 index and OS has already been reported in previous studies, but there are others reporting the opposite (see Section 4.)

No difference in OS was found between patients with low (<80) and patients with high (≥80) KPS (Figure 6c, Table 3). However, for GBM patients with methylated *MGMT* promoter, OS significantly positively correlated with KPS (*r* = 0.55, *p* = 0.044).

Patients who did not receive any primary therapy after surgery had significantly lower OS than those undergoing either radiotherapy (*p* < 0.001) or radio-chemotherapy (*p* < 0.001). No significant difference was found between patients receiving radiotherapy and those who received radio-chemotherapy (*p* = 0.620) (Figure 6d, Table 3). Moreover, no significant difference in OS was found between patients without adjuvant therapy and those with adjuvant therapy (Figure 6e,f, Table 3).

### 3.11. Impact of Age

We found significant differences in *MGMT* promoter and enhancer methylation between patients <60 years and those ≥60 years (Figure 7). In the subgroup of patients with methylated *MGMT* promoter, patients ≥60 years had significantly higher methylation levels of CpGs 72–83 in the *MGMT* promoter than younger ones (Figure 7a).

Significant differences in the methylation status between older and younger patients were also found for the four enhancers (Figure 7b–e). Methylation levels of enhancer 1, enhancer 2, and enhancer 4 were lower in older GBM patients than in younger ones (Figure 7b,c,e). In the case of enhancer 4, significant differences between age subgroups were only found for tumors with methylated *MGMT* promoter (Figure 7e). The association between enhancer 3 methylation and age depended on *MGMT* promoter methylation. In patients with an unmethylated *MGMT* promoter, older patients showed significantly higher methylation levels, whereas in patients with a methylated *MGMT* promoter, older patients showed lower enhancer 3 methylation levels compared to younger patients (Figure 7d).

In our patient cohort, the proliferation index Ki-67 did not correlate with age of the patients (*r* = −0.48, *p* = 0.068). However, patients with Ki-67 ≤ 50% had significantly higher ages than those with Ki-67 > 50% (Figure 8a).

For GBM patients with methylated *MGMT* promoter, we found a significantly negative correlation between KPS and the age of the patients (*r* = −0.64, *p* = 0.010, Figure 8b). In the subgroup of patients with methylated *MGMT* promoter, patients with KPS ≤ 80% were older than those with KPS > 80% (Figure 8c).

Patients <60 years had longer OS compared to those ≥60 years (Figure 8d, Table 3). In samples with methylated *MGMT* promoter, age negatively correlated with OS (*r* = −0.53, *p* = 0.024, Figure 8e).

Our patient cohort contained three long-term survivors (GBM03, GBM04, and GBM23). Long-term survivors are generally defined as GBM patients who survive longer than three years after surgery [23]. We found significant differences in *MGMT* promoter methylation (Appendix A). In addition, the methylation of enhancer 1, enhancer 2, and enhancer 4 was different between long-term survivors and non-long-term survivors of our cohort. For all three enhancers, significant differences were found in case all samples were included in statistical analysis but also for samples with methylated *MGMT* promoter (Appendix A).

### 3.12. Impact of Sex

CpGs 72–83 in the *MGMT* promoter were significantly more highly methylated in female than in male patients (Figure 9a). However, for none of the individual CpGs a significant difference was found (*p* ≥ 0.051).

Significantly higher methylation levels in female patients were also found for enhancer 1 (Figure 9b) and enhancer 2 (Figure 9c). In contrast, several CpGs of enhancer 4 were methylated significantly lower in female than in male patients (Figure 9d).

### 3.13. Two-Way ANOVA Considering Age and Sex

Two-way ANOVA was performed to verify if differences in *MGMT* promoter or enhancer methylation between *TERT* promoter wildtype and mutations C228T and C250T, between *TERT* rs2853669 genotypes, and between patients with low (≤50%) and high (>50%) Ki-67 index found by one-way ANOVA were also significant when considering age (<60 and ≥60 years) or sex.

Interaction effects of *TERT* promoter status with age were found in *MGMT* promoter methylated samples for promoter and enhancer 3 (Appendix A). Few interactions with sex were identified for enhancer 2 and enhancer 3. All significant associations between *MGMT* promoter/enhancer methylation and *TERT* promoter mutations identified by one-way analysis (Section 3.4) were also found in the two-way approach, with the exception of the methylation status of CpG 12–19 of *MGMT* enhancer 1 when considering sex (Appendix A).

An interaction between the *TERT* rs2853669 genotype and age was only found for CpGs 19–22 of enhancer 4 in samples with an unmethylated *MGMT* promoter (Appendix A). An interaction with sex was found for CpGs 72–83 of the *MGMT* promoter in samples not stratified for the *MGMT* promoter methylation status, for CpGs 12–19 of enhancer 1 independent of the *MGMT* promoter methylation status, as well as for CpGs 15–22 of enhancer 3 in samples with an unmethylated *MGMT* promoter. All significant associations between *MGMT* promoter/enhancer methylation and the *TERT* rs2853669 genotype identified by one-way analysis (Section 3.5) were confirmed by two-way ANOVA. However, for a few CpGs, no main effect was found when considering age or sex (Appendix A).

The Ki-67 status was found to interact with age for the methylation levels of CpGs 05–08 and CpGs 37–39 of enhancer 2. An interaction with sex was found for CpGs of enhancer 1 and CpGs 05–08, CpG 08, and CpGs 37–39 of enhancer 2, and exclusively in samples with methylated *MGMT* promoter, CpGs 15–22 of enhancer 3 (Appendix A). In general, Ki-67 status was a main effect when considering age, with the exception of CpGs 12–19 of enhancer 1. No main effect was found for CpGs 72–83 of the *MGMT* promoter when sex was considered (Appendix A).

ANOVA could not be applied for differences between *MGMT* SNP rs16906252 genotypes due to a too-low number of representatives of the TT genotype.

### 3.14. Cell Lines Resulting in Unstable Cell Cultures

Three primary GBM cell lines (GBM36–GBM38) did not result in stable cell cultures (Appendix A). In general, results obtained for these samples were quite different from those obtained for samples resulting in stable cell cultures and were therefore excluded from the statistical analyses described above. All three (100%) samples had an unmethylated *MGMT* promoter, compared to 16 (45.7%) samples resulting in stable cultures (Appendix A). In general, CpGs in enhancers 1–4 had significantly higher in samples GBM36-38 than in samples GBM01–35 (Appendix A). Regarding *MGMT* SNP rs16906252, two (66.6%) samples were heterozygous CT, and one (33.3%) showed the homozygous C wildtype genotype. In the case of GBM01–35, 32 (91.4%) GBM samples were homozygous C, and an (atypical) heterozygous genotype was only determined for one (2.9%) sample. GBM36–38 (100%) showed the *TERT* promoter wildtype, whereas this genotype was only found for three (8.6%) of the samples GBM01–35.

## 4. Discussion

Promoter methylation plays a crucial role in regulating MGMT protein expression in GBMs. However, *MGMT* in all GBMs is not silenced, although the promoter is methylated. Vice versa, some tumors don’t express MGMT despite a demonstrable lack of promoter methylation. These findings suggest that there are further regulatory mechanisms in addition to *MGMT* promoter methylation.

In a very recent study, we investigated if enhancer methylation is one of the mechanisms contributing to *MGMT* regulation [14]. By determining methylation levels of the *MGMT* promoter and four enhancers, we found that the methylation status of several enhancer regions was actually associated with *MGMT* promoter methylation and/or MGMT expression. In samples with an unmethylated *MGMT* promoter, methylation of enhancer 2 correlated with MGMT expression. In addition, the results suggested that methylation of CpGs of enhancer 4 (hs696, [15]), located in intron 2 of the *MGMT* gene, could serve as a potential prognostic biomarker in GBMs [14]. In this study, we investigated if methylation levels of the four enhancers targeted previously are associated with common genetic variants in GBMs, clinical parameters, and demographic characteristics of GBM patients. To date, studies dealing with the association of enhancer methylation with genetic variants, clinical parameters, and/or demographic characteristics in cancer are limited. For *MGMT* in GBM, there are some studies reporting enhancer methylation data, but these studies refer to the cis-acting enhancer element within the *MGMT* promoter [23]. To the best of our knowledge, this is the first paper investigating the methylation of *MGMT* enhancers located outside of the *MGMT* promoter in terms of genetic variants, clinical parameters, and demographic characteristics.

SNP rs16906252, located within the cis-acting enhancer element in the *MGMT* promoter, is one of the genetic variants that has been associated with *MGMT* promoter methylation and MGMT expression [24,25]. The presence of the T allele of rs16906252 was linked to *MGMT* promoter methylation and reduced activity of the *MGMT* promoter. Moreover, among GBM patients with a methylated *MGMT* promoter, carriers of the T allele showed a significant survival benefit compared to patients lacking the T allele [24,25]. In our patient cohort, the prevalence of the rs16906252 T allele was very low. Only 5.7% of the GBM samples were homozygous T, and only 2.9% of the samples showed an (atypical) heterozygous genotype. In contrast to previous studies targeting CpGs 74–78 [24,25], *MGMT* promoter methylation was not associated with the rs16906252 genotype. However, the methylation status of CpGs in *MGMT* enhancers was linked to *MGMT* SNP rs16906252. Methylation of intragenic enhancer 4 (CpGs 02, 03, and 07) was significantly lower in samples of the homozygous T genotype compared to those being homozygous C. In our previous study, we found enhancer 4 methylation negatively correlated with *MGMT* promoter methylation, with CpG 03 showing a particularly strong correlation [14]. Among the three intergenic enhancers, enhancer 2 (mean CpGs 24–27; CpGs 24–27; CpGs 24, 26, and 27) was also significantly lower methylated in homozygous T compared to homozygous C samples. In contrast, enhancer 1 (CpGs 13–15) and enhancer 3 (CpGs 15–22, CpG 16) were significantly more highly methylated in homozygous T compared to homozygous C samples. Our results suggest an association between methylation levels of the enhancers and the *MGMT* SNP rs16906252 genotype. However, due to the low prevalence of the T allele in our patient cohort, further studies are required to confirm our findings.

Mutations in the *TERT* promoter commonly occur in malignant tumors [26,27,28]. *TERT* encodes a rate-limiting catalytic subunit of the enzyme telomerase, which regulates the length of telomeres [29]. Transcriptional activation of *TERT* by mutations plays a crucial role in tumorigenesis [28]. In GBMs, *TERT* is frequently activated by mutation C250T or C228T, located 146 bp and 124 bp upstream of the translation start site (TSS), respectively. Increased telomerase activity, reported for >60–70% of GBM patients, has been associated with poor prognosis [26]. In addition to *TERT* mutations, the SNP rs2853669, located 245 bp upstream of the TSS, has been associated with *TERT* activity. In contrast to mutations C250T and C228T, the less frequent (C) allele of SNP rs2853669 is linked to low *TERT* activity [30,31].

In our patient cohort, *TERT* mutations C228T and C250T were detected in 21 (60.0%) and 11 (31.4%) of the tumors, respectively. In line with the literature [32], the mutations occurred mutually exclusive. Regarding *TERT* SNP rs2853669, 19 (55.9%) GBM samples had the wildtype TT genotype and 15 (44.1%) the heterozygous CT genotype.

Previous findings on associations between *MGMT* promoter methylation, *TERT* promoter mutations, and/or *TERT* SNP rs2853669 are controversial, with some studies reporting associations [33,34], whereas others did not [20,26]. In the subgroup of tumors with methylated *MGMT* promoter, carriers of the C250T mutation showed significantly higher *MGMT* promoter methylation compared to wildtype tumors and C228T mutation carriers. GBM patients, being heterozygous for the SNP rs2853669, had a significantly lower *MGMT* promoter methylation status than those of the TT genotype. Thus, our results are in accordance with previous papers reporting associations between *MGMT* promoter methylation, *TERT* promoter mutations, and *TERT* SNP rs2853669 [33,34]. However, the biological mechanism for these interactions remains to be elucidated.

In addition to *MGMT* promoter methylation, methylation of the three intergenic enhancers but not that of intragenic enhancer 4 was associated with *TERT* mutations. Tumors with *TERT* promoter mutation showed significantly higher methylation of enhancer 1 (C228T, CpGs 12–19) and enhancer 2 (C228T and C250T, CpGs 37–39) compared to tumors lacking TERT mutation. In the subgroup of tumors with methylated *MGMT* promoter, methylation of enhancer 2 (CpG 37, CpG 38, mean methylation of CpGs 37–39) was significantly higher in C228T mutation carriers than in wildtype tumors. In GBM patients with an unmethylated *MGMT* promoter, carriers of C228T mutation showed significantly higher methylation of enhancer 3 (CpGs 15–22) than carriers of C250T mutation. For GBM patients with a methylated *MGMT* promoter, *TERT* C228T mutation was associated with lower methylation of CpGs 15–22 than tumors of wildtype or those with C250T mutation. In the case of *TERT* SNP rs2853669, significant differences between wildtype and SNP genotype were found for all four enhancers investigated. The three intergenic enhancers were more highly methylated in samples of the CT genotype compared to the wildtype TT genotype. In contrast, enhancer 4 was significantly lower methylated in samples of the CT genotype compared to wildtype samples. The difference was more pronounced in samples with an unmethylated *MGMT* promoter. In samples with methylated *MGMT* promoter, a significant difference was found for CpG 09, with heterozygous samples showing lower methylation than wildtype samples.

Ki-67 is a nuclear protein that is expressed in all cell cycle phases except the resting cell phase G0 [35]. Thus, it is widely used as a proliferation marker for human tumor cells [36]. In the context of brain tumors, the Ki-67 index was found to be applicable for differentiating between high and low-grade gliomas, but its relevance as a marker for GBMs is controversially discussed [37]. In this study, the Ki-67 index was specified only for 15 patients; for a further nine patients, the proliferation status of the tumor was available in a semi-quantitative manner. Thirteen (54.2%) patients showed a Ki-67 index >50%, which is rather high compared to other studies, reporting mean Ki-67 values of about 20% [38,39]. Differences in reported Ki-67 values have been mainly attributed to differences in Ki-67 determination, including the use of different antibody clones, antibody formats, and/or staining platforms [40]. In addition, heterogeneity of tumor tissues, e.g., the presence of non-neoplastic cells, activated microglia or macrophages, has an impact on overall proliferative activity [37,41]. Furthermore, differences in Ki-67 data may be caused by interobserver variability [38].

*MGMT* promoter methylation was not significantly correlated with the Ki-67 index. However, we found a significant difference in *MGMT* promoter methylation between tumors with high and low Ki-67 index. Tumors with a high Ki-67 index (>50%) showed significantly higher *MGMT* promoter methylation than tumors with a low (≤50%) Ki-67 index. Due to the high Ki-67 values found for our patient cohort, we set the cut-off at 50%. In other studies, the Ki-67 cut-off was set at lower values, e.g., at 20% [37,39] or 27% [35,42]. Thus, our results are not comparable with previous findings.

The three intergenic enhancers investigated are located between the *Ki-67* gene (upstream) and the *MGMT* gene (downstream). For enhancer 1 (CpG 12, CpG 13, and CpG 18) and enhancer 2 (mean methylation of CpGs 05–08, CpGs 11–14, CpGs 15–18, and CpGs 37–39 as well as ten individual CpGs), the two enhancers being closest to *Ki-67*, we found significantly positive correlation between enhancer methylation and Ki-67 index, but exclusively for GBM samples with an unmethylated *MGMT* promoter. In addition, methylation of enhancer 2 (CpGs 37–39, CpG 08) was significantly higher in GBM patients with a high Ki-67 index (>50%) compared to those with a low (≤50%) Ki-67 index. In samples with unmethylated *MGMT* promoter, we found significant differences for CpGs 05–08, CpGs 24–27, CpGs 37–39, mean of CpGs 37–39, and individual CpG 08 and CpG 37. Methylation of enhancer 1 (CpGs 12–19) was significantly higher in GBM patients with a high Ki-67 index (>50%) compared to those with a low (≤50%) Ki-67 index. Methylation of enhancer 3 (CpGs 15–22) was also significantly higher in patients with high Ki-67 index (Ki-67>50%), but only for GBM samples with methylated *MGMT* promoter. Chen et al., who identified enhancer 2, have already linked enhancer 2 with Ki-67. Deletion of the enhancer region resulted in reduced expression of not only MGMT but also Ki-67, although to a lesser extent [16]. Thus, Chen et al. concluded that *Ki-67* and *MGMT* are in the same topologically associating domain (TAD). TADs are genomic regions with maximal intradomain interactions and minimal interactions with neighboring regions [43,44]. Our results support the hypothesis of Chen et al. In contrast to intergenic enhancers 1, 2, and 3, the methylation status of enhancer 4, located in intron 2 of *MGMT*, was not associated with the Ki-67 index.

By using a cut-off of 8% for *MGMT* promoter methylation, OS was significantly longer for patients with methylated *MGMT* promoter compared to those with an unmethylated *MGMT* promoter, which is in accordance with previous studies [45,46]. For individual CpGs 75, 78, and 80, we found a significant positive correlation with OS [14]. In a recent study by Leske et al., methylation levels of CpGs 72, 74, 75, 81, 83, 85, and 93 in the *MGMT* promoter were found to be strongly associated with the prognosis of GBM patients [23]. In addition to the *MGMT* promoter, we found enhancer 4 to be associated with OS. By setting a cut-off value of 55%, we revealed significantly longer OS for patients with higher enhancer 4 methylation (mean methylation of CpGs 01–03, CpG 03, CpG 07, mean of CpGs 09–13, and CpG 13) compared to those with lower methylation status. In contrast to the *MGMT* promoter and enhancer 4, neither mean methylation levels nor methylation levels of individual CpGs of intergenic enhancers 1–3 were associated with OS of the patients.

Growing evidence suggests that *MGMT* promoter methylation is one of the factors linked to long-term survival, the phenomenon that GBM patients survive longer than three years after surgery [23,47]. In the three long-term survivors of our patient cohort, the *MGMT* promoter was methylated, which is in accordance with the literature. Moreover, we found long-term survival to be associated with methylation of enhancer 1, enhancer 2, and enhancer 4. Since our patient cohort only contained three long-term survivors, further studies are required to confirm our findings.

In our patient cohort, neither *MGMT* promoter nor enhancer methylation correlated with KPS. In terms of survival, only enhancer 4 methylation (CpG 01) negatively correlated with PFS, but exclusively in samples with an unmethylated *MGMT* promoter.

In accordance with previous studies [45], *MGMT* promoter methylation was not associated with the age of the patients at surgery if samples were not stratified by their *MGMT* promoter methylation status. However, in the subgroup of tumors with methylated *MGMT* promoter, patients ≥60 years showed significantly higher *MGMT* promoter methylation than younger ones (<60 years). We also found *MGMT* enhancer methylation to be associated with age. Patients ≥60 years showed significantly lower methylation levels of intragenic enhancer 4 and intergenic enhancers 1 and 2. In the case of enhancer 3, the association with the age of the patients depended on the *MGMT* promoter methylation status. In the subgroup of patients with an unmethylated *MGMT* promoter, younger patients showed significantly lower enhancer 3 methylation levels compared to older ones. For the subgroup of patients with methylated *MGMT* promoter, the opposite finding was observed.

Our patient cohort comprised fourteen (40%) females and 21 (60%) males. We found the *MGMT* promoter was significantly more highly methylated in female patients than in male patients. Sex-specific differences in *MGMT* promoter methylation have already been reported in the literature [48]. Methylation levels of intergenic enhancer 1 (CpGs 12-19, CpG 12, CpG 13, CpG 14, and CpG 16) and enhancer 2 (CpGs 05–08, CpGs 24–27, CpGs 37–39, mean methylation of CpGs 24–27, CpG08, CpGs 25, CpG 26) were also higher in females than in males. In contrast, enhancer 4 methylation (CpGs 07–08, CpGs 09–13, mean methylation CpGs 09–13, CpG 09, CpG 12) was significantly lower in female than in male patients.

Figure 10 summarizes the significant associations between *MGMT* promoter methylation and/or *MGMT* enhancer methylation with MGMT expression, the occurrence of genetic variants, clinical parameters, and demographic characteristics that we found for our patient cohort. With the exception of MGMT protein expression and OS, associations for the *MGMT* promoter were only found when all twelve CpGs were taken into account. We could not identify one enhancer, one enhancer region, nor one individual CpG within the enhancers that was relevant for all parameters and characteristics investigated. The figure clearly indicates the strong impact of the *MGMT* promoter methylation status on the associations found. For example, associations between enhancer 2 methylation and MGMT protein expression and the Ki-67 index were predominantly found for samples with an unmethylated *MGMT* promoter and those with *TERT* mutations, mainly for samples with methylated *MGMT* promoter. In the case of enhancer 4, associations with the *TERT* SNP genotype and age were strongly influenced by the *MGMT* promoter methylation status.

We also found associations between various clinical parameters for our patient cohort, with most of them being in accordance with previous studies. Patients who did not receive any primary therapy after surgery had significantly lower OS than those undergoing either radiotherapy or radio-chemotherapy, which is in line with the literature [12,48]. OS was longer in patients undergoing adjuvant therapy with TMZ alone than in those without adjuvant therapy. In a number of studies, higher age has been associated with worse clinical outcomes in GBM [8,47]. This association was also observed in our study.

For our patient cohort, we did not find a significant association between sex and OS, which is also in accordance with previous studies [8,35,49].

Surprisingly, Ki-67 levels positively correlated with OS for the 15 GBM patients for which quantitative Ki-67 data were available. In addition, GBM patients with a Ki-67 index >50% had significantly longer OS compared to patients with a low (≤50%) Ki-67 index. One would obviously expect that higher proliferation activity of the tumor is associated with a worse survival prognosis. In the literature, findings on Ki-67 and OS are controversial. A few studies also linked higher Ki-67 values to longer OS in GBM [42,50]. However, the majority of studies report a negative association between Ki-67 and OS [39,49,51,52]. There are also studies that did not find the Ki-67 index to be linked to OS [53]. Controversial discussion on associations between Ki-67 and OS is not limited to GBM but is also ongoing for other cancer types, e.g., melanoma [54] and breast cancer [55]. Longer OS of patients with higher Ki-67 index could be explained by the fact that highly proliferating tumors are particularly prone to cytotoxic effects of chemotherapy or radio-chemotherapy [42].

Growing evidence suggests that radio chemotherapy improves the treatment efficacy for GBM patients compared to radiotherapy alone [56]. However, for our patient cohort, we did not find a significant difference in OS between patients receiving radiotherapy and those who received radio-chemotherapy. Most likely, the subgroup receiving radiotherapy alone was too small in our patient cohort.

Our original patient cohort contained three patients for which stable cell cultures could not be obtained. These samples were excluded from statistical analysis because they differed in many data from samples for which stable cell cultures were achieved. Among others, all three samples showed the *TERT* promoter wildtype, which could explain why cells could not achieve immortality. However, for other cells showing the *TERT* promoter wildtype, stable cell cultures could be achieved, suggesting that additional factors were involved.

## 5. Conclusions

Our results show that enhancer methylation does not only play a crucial role in regulating gene expression. The findings clearly indicate that methylation of the enhancers investigated is associated with genetic variants, clinical parameters, and demographic characteristics in GBM.

In general, the associations identified for intragenic enhancer 4, located in intron 2 of *MGMT*, were quite different from those found for the three intergenic enhancers. Intragenic enhancer 4 was the only enhancer whose methylation levels were negatively correlated with *MGMT* promoter methylation, as shown previously. In addition, an association of the methylation status with OS was exclusively found for enhancer 4. Thus, enhancer 4 methylation could be a potential prognostic biomarker for GBM. In contrast to the three intergenic enhancers, enhancer 4 methylation was neither associated with *TERT* mutations nor with the Ki-67 index. The methylation status of enhancer 4 was significantly lower in female than in male patients, whereas for the three intergenic enhancers, the opposite finding was observed.

Among the three intergenic enhancers, enhancer 2 methylation could be a potential biomarker in GBM. Methylation of several CpGs of enhancer 2 was positively correlated with MGMT expression, as shown previously and/or the Ki-67 index, as shown in this work. Both associations were observed for the subgroup of samples with an unmethylated *MGMT* promoter. Associations with genetic variants, clinical parameters, and demographic characteristics were also observed for enhancer 1 and enhancer 3.

In general, the associations with genetic variants, clinical parameters, and demographic characteristics were caused by enhancer regions rather than by individual CpGs, hinting at a complex interplay of multiple CpGs and enhancer regions.

Some of our findings were obtained without stratifying samples by their *MGMT* promoter methylation status. However, other associations were restricted to subgroups, either samples with methylated or samples with an unmethylated *MGMT* promoter, underpinning the relevance of the *MGMT* promoter status in GBMs.

We would like to stress that our findings are based on a rather heterogenous sample set, especially regarding primary and adjuvant therapy of the patients. Discrepancies in treatment are expected to have an impact on PFS and OS and thus may confound the robustness of the associations between *MGMT* methylation levels and these clinical parameters. Further studies are required to confirm our findings.

## Figures and Tables

**Figure 1 cancers-15-05777-f001:**
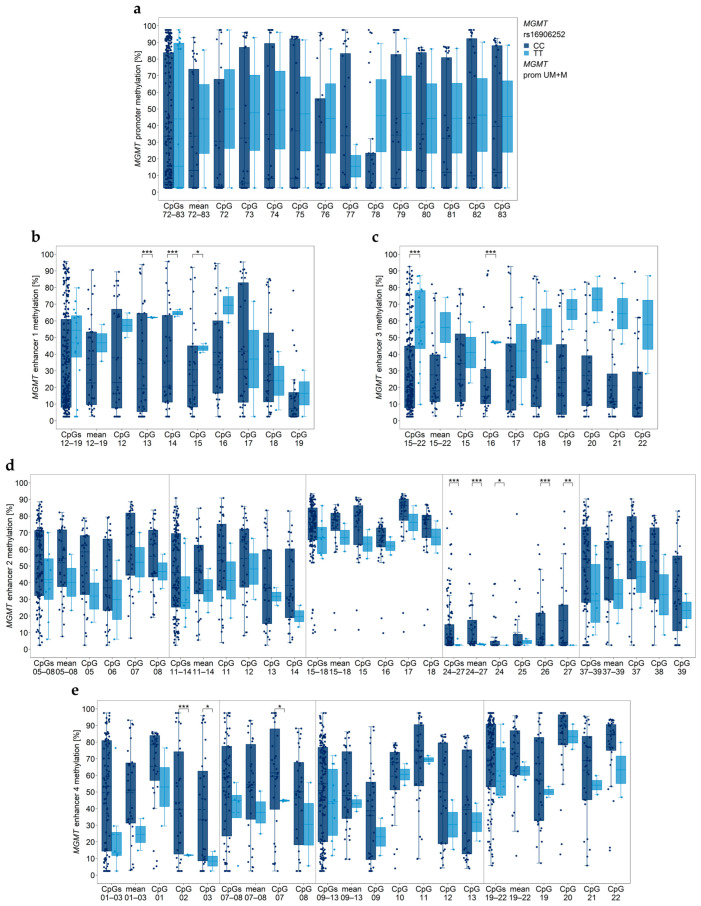
Association between *MGMT* promoter/enhancer methylation and *MGMT* SNP rs16906252 genotypes. *MGMT* (**a**) promoter, (**b**) enhancer 1, (**c**) enhancer 3, (**d**) enhancer 2, and (**e**) enhancer 4 regions. Gray vertical lines separate data obtained with different assays. Significance levels: * *p* ≤ 0.05, ** *p* ≤ 0.01, *** *p* ≤ 0.001. Data points represent the mean of two independent PCR-PSQ runs.

**Figure 2 cancers-15-05777-f002:**
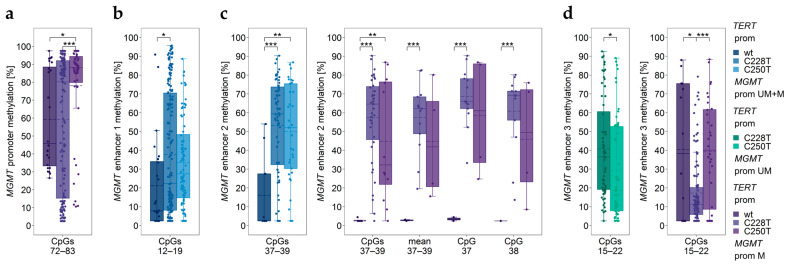
Significantly different *MGMT* promoter/enhancer methylation levels between *TERT* promoter (prom) wildtype (wt) and mutations C228T and C250T. *MGMT* (**a**) promoter, (**b**) enhancer 1, (**c**) enhancer 2, and (**d**) enhancer 3 regions. prom M—promoter methylated (purple), UM—unmethylated (green), UM + M (blue) patients. GBM18 was the only sample with an unmethylated *MGMT* promoter lacking C228T and C250T mutation; it was not included in statistical analysis. Significance levels—* *p* ≤ 0.05, ** *p* ≤ 0.01, *** *p* ≤ 0.001. Data points represent the mean of two independent PCR-PSQ runs.

**Figure 3 cancers-15-05777-f003:**
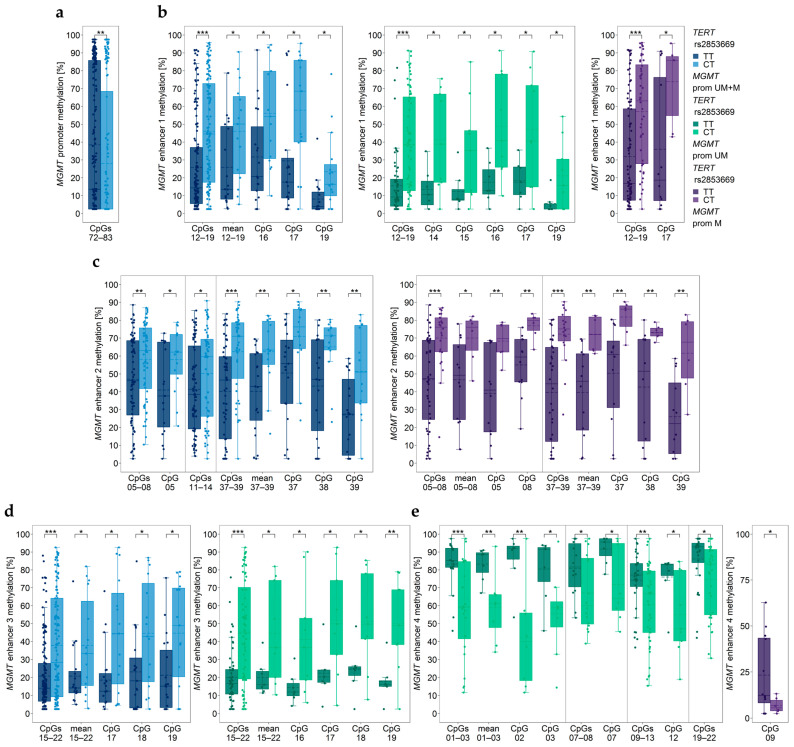
Significantly different *MGMT* promoter/enhancer methylation levels between *TERT* rs2853669 genotypes. *MGMT* (**a**) promoter, (**b**) enhancer 1, (**c**) enhancer 2, (**d**) enhancer 3, and (**e**) enhancer 4 regions. prom M—promoter methylated (purple), UM—unmethylated (green), UM + M (blue) patients. Significance levels: * *p* ≤ 0.05, ** *p* ≤ 0.01, *** *p* ≤ 0.001. Data points represent the mean of two independent PCR-PSQ runs.

**Figure 4 cancers-15-05777-f004:**
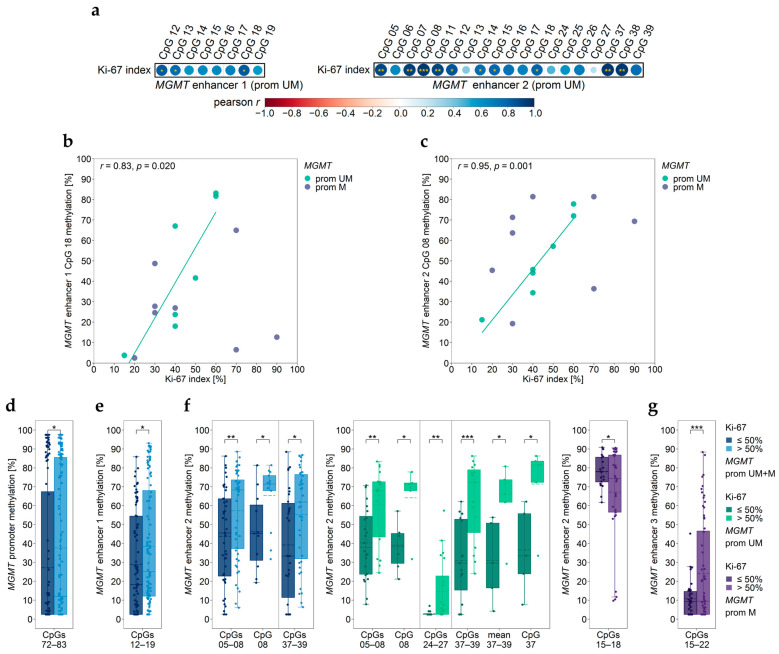
Association of *MGMT* promoter/enhancer methylation with Ki-67 index. (**a**–**c**) Correlation plots and two exemplary scatterplots for *MGMT* (**b**) enhancer 1 (CpG18) and (**c**) enhancer 2 (CpG08) indicating correlations in patients with an unmethylated *MGMT* promoter. (**d**–**g**) Significant differences between patients with low (≤50%) and high (>50%) Ki-67 index for *MGMT* (**d**) promoter, (**e**) enhancer 1, (**f**) enhancer 2, and (**g**) enhancer 3 regions. Pearson’s correlation coefficients are shown in color ranges of dark red (−1.0)–dark blue (1.0); point size (0–±1.0), significant coefficients are highlighted in yellow. Prom M—promoter methylated (purple), UM—unmethylated (green), UM + M (blue) patients. Significance levels: * *p* ≤ 0.05, ** *p* ≤ 0.01, *** *p* ≤ 0.001. Data points represent the mean of two independent PCR-PSQ runs.

**Figure 5 cancers-15-05777-f005:**
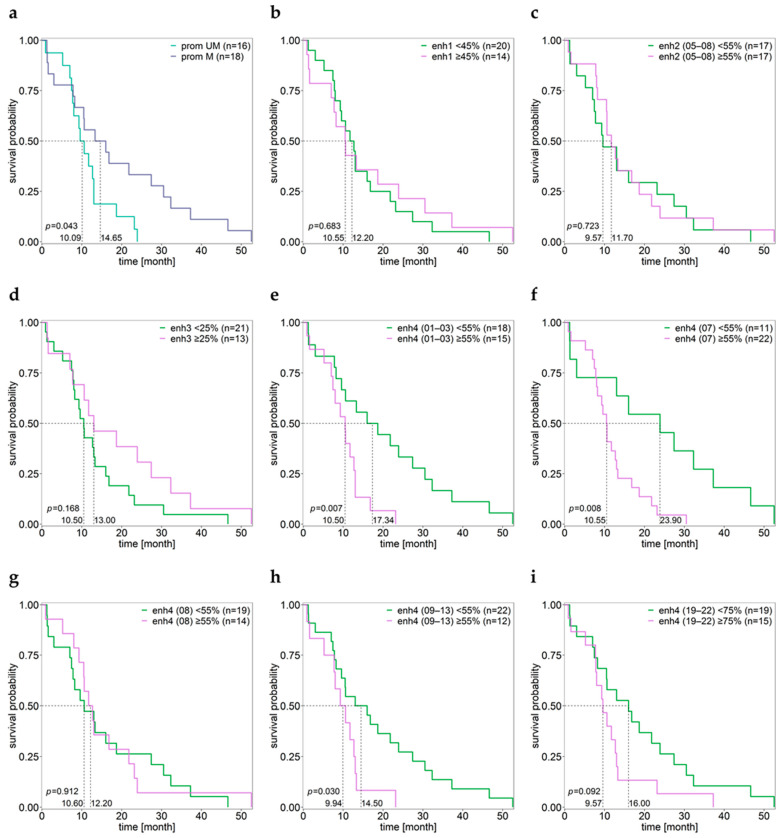
Kaplan–Meier survival analysis. Samples were grouped by the mean methylation status of *MGMT* (**a**) promoter CpGs 72–83, (**b**) enhancer 1 CpGs 12–19, (**c**) enhancer 2 CpGs 05–08, (**d**) enhancer 3 CpGs 15–22, and (**e**–**i**) enhancer 4 for CpGs (**e**) 01–03, (**f**) 07, (**g**) 08, (**h**) 09–13, and (**i**) 19–22. enh—enhancer, prom—promoter. Patient GBM27 was excluded due to a lack of OS data.

**Figure 6 cancers-15-05777-f006:**
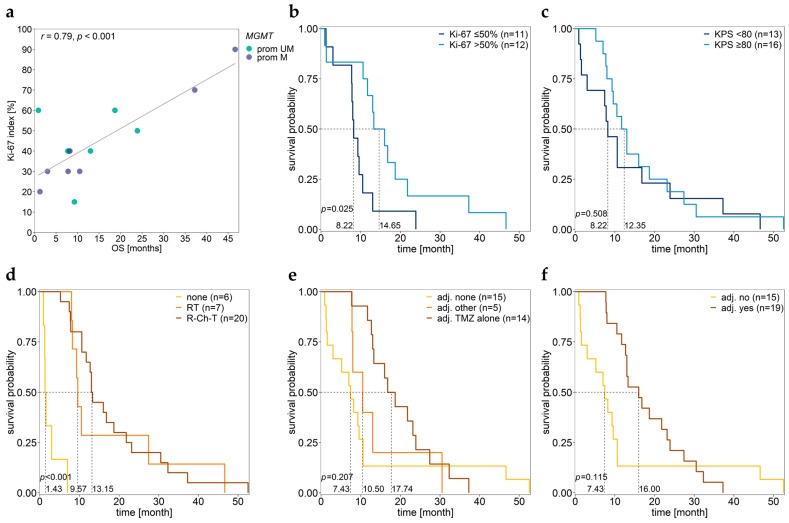
Association of Ki-67 index, KPS, and therapy with OS. Scatterplot showing the (**a**) correlation between Ki-67 index and OS in patients independent of their *MGMT* promoter methylation status. (**b**–**f**) Kaplan–Meier survival analysis of samples grouped by (**b**) Ki-67 index, (**c**) KPS, (**d**) primary therapy, and (**e**,**f**) adjuvant therapy status. Patient GBM27 was excluded due to a lack of OS data. Prom M—promoter methylated (purple), UM—unmethylated (green), UM + M (gray) samples.

**Figure 7 cancers-15-05777-f007:**
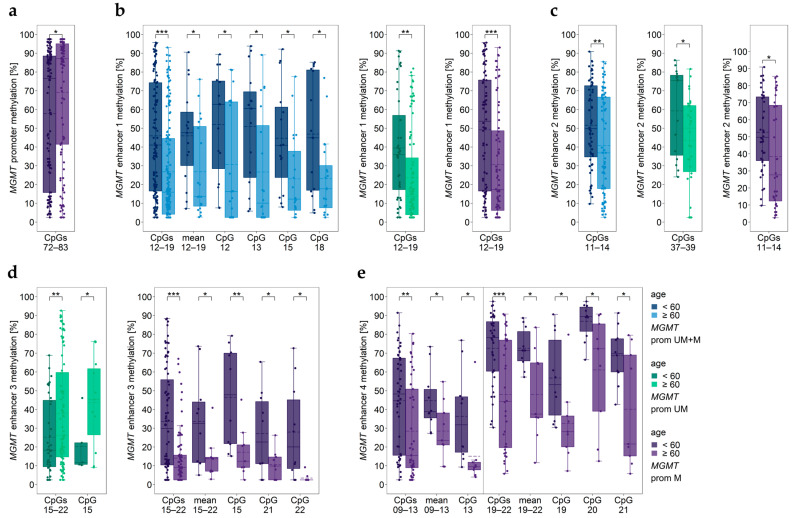
Significantly different *MGMT* promoter/enhancer methylation levels between younger (<60 years) and older (≥60 years) patients. *MGMT* (**a**) promoter, **(b**) enhancer 1, (**c**) enhancer 2, (**d**) enhancer 3, and (**e**) enhancer 4 regions. prom M—promoter methylated (purple), UM—unmethylated (green), UM + M (blue) samples. Significance levels: * *p* ≤ 0.05, ** *p* ≤ 0.01, *** *p* ≤ 0.001. Data points represent the mean of two independent PCR-PSQ runs.

**Figure 8 cancers-15-05777-f008:**
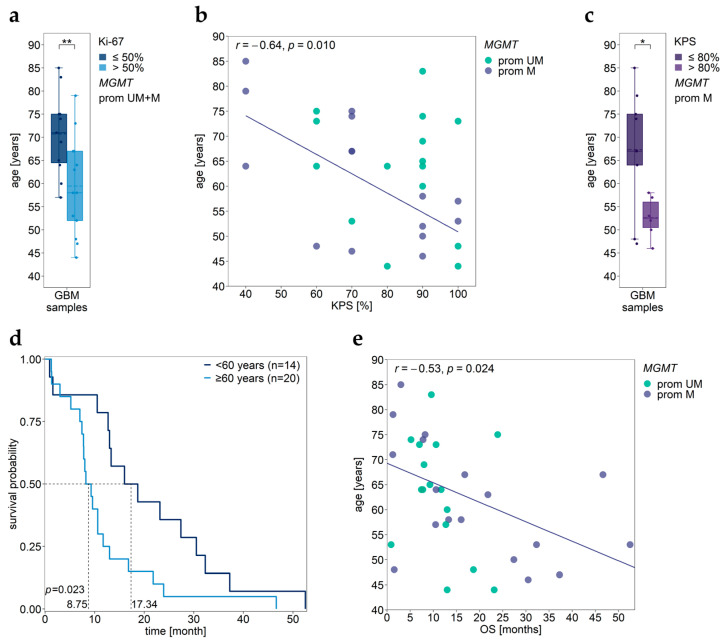
Association of age with Ki-67 index, KPS, and OS. (**a**) Significant difference in age between patients with low (≤50%) and high (>50%) Ki-67 index. (**b**) Correlation between age and KPS in patients with methylated *MGMT* promoter. (**c**) Significant difference in age of patients with low (≤80%) and high (>80%) KPS. (**d**) Kaplan–Meier survival analysis of patients grouped by age (<60 years, ≥60 years). Patient GBM27 was excluded from the survival analysis due to a lack of OS data. (**e**) Correlation between age and OS in patients with methylated *MGMT* promoter. Significance levels: * *p* ≤ 0.05, ** *p* ≤ 0.01. Prom M—promoter methylated (purple), UM—unmethylated (green), UM + M (gray) patients.

**Figure 9 cancers-15-05777-f009:**
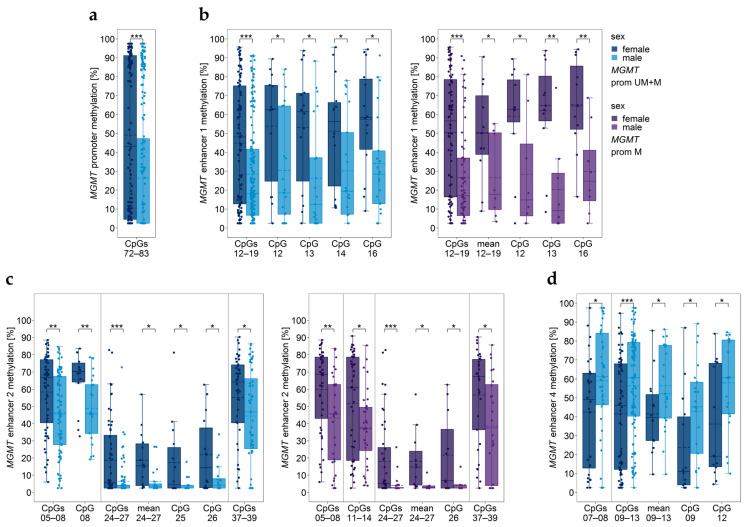
Significantly different *MGMT* promoter/enhancer methylation levels between female and male patients. *MGMT* (**a**) promoter, (**b**) enhancer 1, (**c**) enhancer 2, and (**d**) enhancer 4 regions. prom M—promoter methylated (purple), UM—unmethylated, UM + M (blue) samples. Significance levels: * *p* ≤ 0.05, ** *p* ≤ 0.01, *** *p* ≤ 0.001. Data points represent the mean of two independent PCR-PSQ runs.

**Figure 10 cancers-15-05777-f010:**
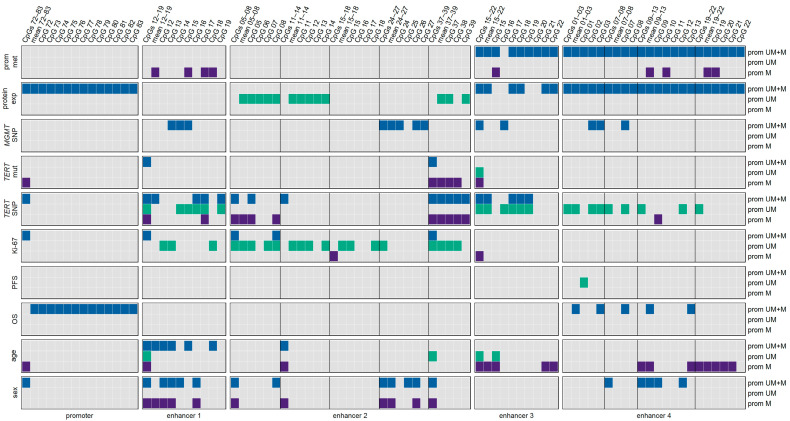
Summary of associations found for *MGMT* promoter and *MGMT* enhancer methylation. Significant associations are highlighted for prom M—promoter methylated (purple), UM—unmethylated (green), and UM + M (blue) samples.

**Table 1 cancers-15-05777-t001:** Clinical and genetic data of GBM patients for which stable cell cultures could be established.

Patient	Age [year]	Sex	KPS [%]	Ki-67 [%]	Primary Therapy	Adjuvant Therapy	OS [m]	PFS [m]	MGMT Exp	*MGMT* rs16906252	*TERT* Prom	*TERT* rs2853669
GBM01	64	f	60	n.s.	R-Ch-T	none	7.43	4.00	0.00	CC	C250T	TT
GBM02	85	m	40	≤50 (30)	none	none	3.00	n.s.	0.00	CC	wt	TT
GBM03	53	f	100	n.s.	R-Ch-T	none	52.50	4.50	0.00	CC	C228T	CT
GBM04	67	f	70	>50 (90)	RT	none	46.63	0.53	0.00	CC	C228T	TT
GBM05	57	f	100	≤50 (30)	RT	CCNU	10.50	6.00	0.00	CC	C228T	TT
GBM06	46	m	90	n.s.	R-Ch-T	TMZ + CCNU	30.50	4.00	0.00	CT (atyp.)	C228T	TT
GBM07	50	f	90	n.s.	RT	TMZ	27.40	n.s.	0.00	CC	C250T	n.s.
GBM08	74	f	70	≤50 (30)	R-Ch-T ^1^	Avastin + TMZ	7.79	n.s.	0.00	CC	C228T	CT
GBM09	48	m	60	n.s.	none	none	1.55	n.s.	0.00	CC	wt	TT
GBM10	64	m	80	>50	R-Ch-T	TMZ	11.70	n.s.	0.20	CC	C228T	CT
GBM11	73	f	60	>50	R-Ch-T	no	10.60	n.s.	0.16	CC	C228T	TT
GBM12	44	m	80	>50	R-Ch-T	TMZ	13.00	9.00	1.10	CC	C250T	CT
GBM13	65	m	90	≤50 (15)	RT ^1^	none	9.27	n.s.	1.20	CC	C228T	TT
GBM14	69	m	90	≤50 (40)	RT	TMZ + CCNU	8.00	n.s.	0.04	CC	C250T	TT
GBM15	73	f	100	n.s.	none	none	7.00	2.63	1.10	CC	C250T	CT
GBM16	83	m	90	≤50	RT	none	9.57	9.00	1.00	CC	C228T	CT
GBM17	74	m	90	n.s.	R-Ch-T	none	5.19	n.s.	1.32	CC	C250T	CT
GBM18	44	m	100	n.s.	R-Ch-T	TMZ	23.15	n.s.	0.25	CC	wt	TT
GBM19	48	m	100	>50 (60)	R-Ch-T	TMZ	18.67	4.54	0.09	CC	C228T	CT
GBM20	75	m	60	≤50 (50)	Ch-T	TMZ	23.90	5.98	0.06	TT	C228T	CT
GBM21	60	m	90	≤50 (40)	R-Ch-T	Avastin	13.00	3.00	0.40	CC	C228T	TT
GBM22	53	m	70	>50 (60)	none	none	0.89	n.s.	0.54	CC	C250T	TT
GBM23	47	f	70	>50 (70)	R-Ch-T	TMZ	37.25	8.00	0.00	CC	C250T	CT
GBM24	64	m	90	≤50 (40)	R-Ch-T	TMZ	7.73	n.s.	0.49	CC	C228T	CT
GBM25	67	f	70	>50	R-Ch-T	TMZ	16.80	10.00	0.00	CC	C228T	TT
GBM26	75	f	70	≤50 (40)	RT	none	8.22	n.s.	0.00	CC	C228T	CT
GBM27	52	f	90	>50 (70)	R-Ch-T	TMZ	n.s.	n.s.	0.00	TT	C250T	TT
GBM28	63	m	n.s.	>50	R-Ch-T	TMZ	21.80	8.00	0.00	CC	C250T	CT
GBM29	79	f	40	>50	none	none	1.31	n.s.	0.00	CC	C228T	CT
GBM30	58	m	90	>50	R-Ch-T	TMZ	16.00	4.31	0.00	CC	C228T	TT
GBM31	58	f	n.s.	>50	R-Ch-T	TMZ	13.30	4.00	0.00	CC	C228T	TT
GBM32	71	m	n.s.	≤50 (20)	none	none	1.25	n.s.	0.00	CC	C228T	TT
GBM33	57	m	n.s.	n.s.	R-Ch-T	TMZ	12.70	6.00	0.20	CC	C250T	CT
GBM34	53	m	n.s.	n.s.	R-Ch-T	TMZ	32.30	22.00	0.00	CC	C228T	TT
GBM35	64	m	40	n.s.	R-Ch-T	none	10.60	n.s.	0.00	CC	C228T	TT

GBM01–35—glioblastoma multiforme patients. Applied drugs were bevacizumab (Avastin), Lomustine (CCNU—(1-(2-chloroethyl)-3-cyclohexyl-1-nitrosourea)), and temozolomide (TMZ). atyp—atypical HRM-PSQ result [21], Ch-T—chemotherapy, exp—expression, f—female, KPS—Karnofsky Performance Score, m—male, mut—mutation, n.s.—not specified, OS—overall survival, PFS—progression-free survival, R-Ch-T—radio-chemotherapy, RT—radiotherapy, wt—wildtype. ^1^ Therapy had to be discontinued.

**Table 2 cancers-15-05777-t002:** Cox proportional hazards models for *MGMT* promoter and enhancer 4 regions.

Promoter				Enhancer 4			
Cut-Off	CpG	Hazard Ratio (95% CI)	*p*-Value	Cut-Off	CpG	Hazard Ratio (95% CI)	*p*-Value
(<8% vs. ≥8%)	mean 72–83	2.18 (1.01–4.71)	**0.042**	(≥55% vs. <55%)	mean 01–03	2.97 (1.31–6.74)	**0.007**
	72	2.81 (1.23–6.40)	**0.011**		01	0.71 (0.31–1.60)	0.402
	73	2.55(1.15–5.66)	**0.018**		02	2.05 (0.97–4.30)	0.054
	74	2.52 (1.16–5.50)	**0.017**		03	2.27 (1.01–5.10)	**0.042**
	75	2.18(1.01–4.71)	**0.042**		mean 07–08	1.81 (0.25–1.12)	0.118
	76	2.76 (1.24–6.14)	**0.010**		07	3.27 (1.32–8.12)	**0.008**
	77	2.76 (1.24–6.14)	**0.010**		08	0.96 (0.47–1.97)	0.915
	78	3.35 (1.43–7.85)	**0.004**		mean 09–13	2.34 (1.06–5.13)	**0.030**
	79	2.36 (1.08–5.17)	**0.028**		09	2.19 (0.94–5.08)	0.063
	80	2.18 (1.01–4.71)	**0.042**		10	1.93 (0.87–4.27)	0.102
	81	2.18 (1.01–4.71)	**0.042**		11	1.83 (0.76–4.37)	0.172
	82	2.36 (1.08–5.17)	**0.028**		12	1.18 (0.59–2.35)	0.641
	83	2.18 (1.01–4.71)	**0.042**		13	2.30 (1.05–5.07)	**0.033**
				(≥75% vs. <75%)	mean 19–22	1.84 (0.90–3.75)	0.092
					19	1.43 (0.69–3.00)	0.337
					20	1.22 (0.52–2.83)	0.647
					21	1.11 (0.56–2.23)	0.762
					22	0.85 (0.38–1.91)	0.696

bold—significant (*p* ≤ 0.05), underlined—Hazard Ratio given for this group.

**Table 3 cancers-15-05777-t003:** Univariate Cox proportional hazards model.

Covariate	Hazard Ratio (95% CI)	*p*-Value
Ki-67 index (≤50% vs. >50%)	2.72 (1.11–6.70)	**0.024**
KPS (≥80% vs. <80%)	0.78 (0.37–1.64)	0.507
primary therapy (R-Ch-T vs. none or RT)	0.01 (0.001–0.11)	**<0.001**
adjuvant therapy (TMZ alone vs. none or other)	0.50 (0.23–1.09)	0.201
adjuvant therapy (yes vs. no)	0.56 (0.27–1.16)	0.113
age (≥60 vs. <60)	2.28 (1.11–4.70)	**0.022**

bold—significant (*p* ≤ 0.05), underlined—Hazard Ratio given for this group.

## Data Availability

The datasets generated during the current study are available from the corresponding author on reasonable request.

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
