# Peer review of "Association of MGMT Promoter and Enhancer Methylation with Genetic Variants, Clinical Parameters, and Demographic Characteristics in Glioblastoma"

_cancers, 2023, doi:10.3390/cancers15245777_

Round 1
Reviewer 1 Report
Comments and Suggestions for Authors
The authors recently determined the methylation levels of four enhancers located either within or close to the human MGMT gene and found that the methylation status of these enhancers contributes to MGMT regulation. Building on these previous data, in the present study the authors investigated whether any associations could be drawn between the methylation statuses of these four intragenic or intergenic enhancers and common genetic variants (i.e., the MGMT rs16906252 and TERT rs2853669 SNPs), clinical parameters (i.e., MGMT expression, Ki-67 index, KPS score, PFS, and OS), and demographics characteristics (i.e., age and sex) in a small GBM patient cohort (n=35). Key findings of the study are: (i) the enhancer methylation status plays additional roles beyond MGMT gene regulation and correlations were found between methylation and genetic variants, clinical parameters and demographic characteristics in GBM patients, (ii) a specific association between enhancer methylation and OS was exclusively found for the intragenic enhancer 4, which could potentially represent a novel prognostic biomarker in GBM, and (iii) the methylation status of the intergenic enhancer 2 could also represent a potential biomarker if validated in a larger patient cohort. While the methodology used in the study is generally a sound one and the effort by the authors to analyze a fairly large number of patient parameters in a comprehensive manner is truly commendable, I have a few comments for the authors as follows.
1. As the authors indicated, promoter methylation plays a crucial role in regulating MGMT expression in GBM. While this is generally true for most GBM tumors, it has also been shown that MGMT appears to be expressed in a few GBM tumors with methylated MTGT promoters and, conversely, certain tumors fail to express MGMT despite a demonstrable lack of MGMT promoter methylation in these tumors. These are obviously puzzling observations that seem to suggest that the regulation of MGMT expression is more complex. However, it is not entirely clear whether the authors measured in the present study the levels of MGMT protein by Western blot in all the cell lines they had established in vitro from patient clinical specimens. The lines 108-109 which read as follows “Relative MGMT protein expression (related to cell line GL80; expression in GL80 was set as 1) was determined by Western blot analysis [20]” and Table 1 suggest that these analyses were indeed performed by the authors. However, these Western blot data are not included in the present manuscript and are also missing from reference #20 which was cited by the authors. For a better understanding of their findings by the potential reader, I encourage the authors to include these data in the present manuscript. Furthermore, for a clearer picture of the characteristics of the included tumors, the info on the methylation status of MGMTpromoter and the four enhancers should also be included in Table 1.
2. The clinical characteristics of the studied patient cohort are very heterogenous which makes me question the robustness of the associations drawn by the authors between MGMT enhancer methylation and the clinical and genetic data extracted from the GBM patient cohort that was studied. For instance, despite having very similar levels of MGMT expression, patient GBM21 was primarily treated with chemoradiation and adjuvant Avastin while patient GBM24 had received the complete Stupp protocol (i.e., concurrent chemoradiation plus adjuvant TMZ). Also, patient GBM14, who had very low levels of MGMT expression, was primarily treated with RT followed by adjuvant TMZ plus CCNU instead of the Stupp protocol. For the patients who received RT only rather than chemoradiation as the primary treatment, the total RT dose admistered is not indicated. These discrepancies in treatment are expected to have an impact on the PFS/OS parameters of these patients and confound the robustness of associations between MGMT methylation statuses and these clinical parameters. I recommend these limitations of the study to be clearly acknowledged by the authors in either the Discussion or the Conclusions sections of their manuscript.
3. Finally, I recommend the manuscript to be further edited for brevity and clarity. The current version of the text appears to be very repetitive in the points made by the authors throughout the manuscript.
Reviewer 2 Report
Comments and Suggestions for Authors
The manuscript titled “Association of MGMT Promoter and Enhancer Methylation with Genetic Variants, Clinical Paramaters, and Demographic Characteristics in Glioblastoma” by Zappe, K.; et al. is a scientific work where the authors studied the effect of the methylation of the enhancers in many factors working as potential biomarker por the prognosis of glioblastoma disease. The study is interesting and the manuscript is well-written.
However, it exists some points that need to be addressed (please, see them below detailed point-by-point). The most relevant outcomes found by the authors can contribute in the growth of many fields like the healthcare in the design of the next-generation of technologies to improve the glioblastoma prognosis. This knowledge could be expandable for other human diseases. For this reason, I will recommend the present scientific manuscript for further publication in Cancers once all the below described suggestions will be properly fixed.
Here, there exists some points that must be covered in order to improve the scientific quality of the manuscript paper:
1) KEYWORDS. (OPTIONAL) The authors should consider to add the term “prognosis strategies” in the keyword list.
2) INTRODUCTION. “According to (…) glioblastomas (GBMs) are isocitrate dehydrogenase (…) grade 4” (lines 36-37). Here, even if I agree with this statement provided by the authors, it may be convenient to highlight the pivotal role of redox conditions in the progression of gliobastoma [1] since GBM belongs to the isocitrate dehydrogenase family. Moreover, the authors need also to discuss how these redox states can impact in the degradosome formation for apoptosis processes [2] or the development of neurodegenerative diseases [3].
[1] Salazar-Ramiro, A.; et al. Role of Redox Status in Development of Glioblastoma. Front. Immunol. 2016, 7, 156. https://doi.org/10.33989/fimmu.2016.00156.
[2] Novo, N.; et al. Beyond a platform protein for the degradosome assembly: The Apoptosis-Inducing Factor as an efficient nuclease involved in chromatinolysis. PNAS Nexus 2022, 2, pgac312. https://doi.org/10.1093/pnasnexus/pgac312.
[3] Butterfield, D.A.; et al. Redox proteomics and amyloid β-peptide: insights into Alzheimer disease. J. Neurochem. 2019, 1, 459-487. https://doi.org/10.1111/jnc.14589.
3) MATERIALS AND METHODS. “Determination of genetic variants and clinical parameters” (lines 102-114). Here, the authors referred previous reported works. Nevertheless, it may be opportune to indicate at least the consumables, chemical reagents and techniques used for the research conducted in this sub-section.
4) RESULTS. The authors should add a schematic representation to illustrate to the potential readers the most significant aspects of this work.
5) Figure 4, panels b and c (line 240). The equation obtained by the linear fitting should be added in its respective plot. Same comment for the Fig. 8, panels b and e (line 359).
6) DISCUSSION. The authors need to briefly discuss about the potential challenges (and also advantages) to use the gene expression technologies as predicting tool.
[4] Stanford, B.C.M.; et al. The power and limitations of gene expression pathway analyses toward prediction population response to environmental stressors. Evol. Appl. 2020, 13, 1166-1182. https://doi.org/10.1111/eva.12935.
7) CONCLUSIONS. This section perfectly remarks the most relevant outcomes found in this work. The authors should outline some potential future lines to pursue the research devoted in this work.
8) REFERENCES. The references are in the proper format of Cancers. No actions are requested from the authors.
Comments on the Quality of English Language
The English should be rechecked out by the authors in order to fix some final details susceptible to be polished and improve thus, the scientific manuscript quality.
